# Cooperative Localization Approach for Multi-Robot Systems Based on State Estimation Error Compensation

**DOI:** 10.3390/s19183842

**Published:** 2019-09-05

**Authors:** Shijie Zhang, Yi Cao

**Affiliations:** College of Electrical Engineering, Henan University of Technology, Zhengzhou 450052, China; caoyioffice@163.com

**Keywords:** cooperative localization, multi-robots system, communication delays, state estimation error compensation

## Abstract

In order to improve the localization accuracy of multi-robot systems, a cooperative localization approach with communication delays was proposed in this paper. In the proposed method, the reason for the time delay of the robots’ cooperative localization approach was analyzed first, and then the state equation and measure equation were reconstructed by introducing the communication delays into the states and measurements. Furthermore, the cooperative localization algorithm using the extended Kalman filtering technique based on state estimation error compensation was proposed to reduce the state estimation error of delay filtering. Finally, the simulation and experiment results demonstrated that the proposed algorithm can achieve good performance in location in the presence of communication delay while having reduced computational and communicative cost.

## 1. Introduction

In recent years, multi-robot systems have received widespread attention as a team of robots could increase reliability and performance when compared to an individual robot. In addition, localization accuracy can be improved by teamwork through using shared information. At present, multi-robot systems have been applied in many fields such as target tracking [1,2], data collection [3], rescue [4,5,6], and formation [7,8].

Cooperative work is the greatest advantage of multi-robot systems, and precise positioning is the basis for completing cooperative work. Positioning methods are an important research direction of multi-robot systems. The Global Positioning System (GPS)is one way to localize robots in multi-robot systems, however, the GPS signals are not easily received in the room, and the direction information cannot be obtained directly. The cooperative localization (CL) method is an alternative approach where each robot infers its own position according to the position of other robots in a multi-robot system [9]. Since the previous works of Kurazume et al. [10] in 1994, the cooperative localization problem of multi-robot systems has attracted the interest of many researchers in the past few years [11,12].

A cooperative localization method based on the extended Kalman filter (EKF) was proposed in [13], where the position and posture information of each robot measured by the sensor was updated in time, and the entropic criterion was utilized to ensure that the optimal measurement was used to reduce the uncertainty of the robot pose estimation. In [14], to reduce the influence of uncertainty on the accuracy of CL, the upper bound of uncertainty was described as the characteristic function of the robot sensor time and noise. In this case, the CL algorithm ignores the influence of uncertainty in state propagation and relative position measurement, thereby the computational efficiency is improved. A sample-based Monte Carlo localization (MCL) algorithm was introduced in [15] to improve the accuracy of the cooperative positioning of multi-robot systems. The MCL algorithm was extended in [16] to the case of two robots, if a map of the environment is available to both robots. However, the drawback of the MCL approach is that it can only be used in known environments.

Vision-based navigation technology is also used in cooperative positioning. In order to deal with uncertainty and external disturbance in the measurement, Kalman filters were also employed in [17,18,19] to estimate the position and orientation, and improve the measurement accuracy and robustness of the system.

In practical application, the communication constraints should be taken into account, the information from different types of sensors in the multi-robot systems need to be fused. However, due to the different working frequency and the different information processing time of these different proprioceptive sensors, it causes many problems such as time-delay, which means that delayed measurements are used when state filtering the sensor measurement outputs. Unexpected positioning errors may increase or even lead to filtering divergence if the influence of the communication delay is ignored. In order to solve this problem, many methods have been proposed, and the commonly used method to solve information delay is to use robust theory to predict and compensate the random delay model. These methods are generally used in complex network systems with more sensors, shorter time delay, and disorderly measurement [20,21,22,23,24,25,26,27,28]. A delayed extended Kalman filter (DEKF) method was proposed to solve the communication delay problem in CL [29,30], The method proposed in [29,30] first augmented all Autonomous Underwater Vehicle(AUV) state quantities to each AUV’s positioning filter, and then the state quantity with time delay was subjected to delay processing for filtering estimation. However, this method has a large amount of calculation and requires a lot of time for the measurement information.

Considering the reasons above, to compensate for the positioning error caused by communication delay in cooperative positioning, an extended Kalman filtering (EKF) based on state estimation error compensation was designed in this paper to deal with the communication delay. In the proposed method, the delayed measurements are predicted in advance to compensate the state estimation error; the time delay is introduced into the reconstructed system state and measurement equation; and then the delayed extended Kalman filter based on measurement update is designed. The proposed cooperative localization algorithm is applied to the leader–follower robot formation to demonstrate the effectiveness of the proposed strategy.

## 2. Problem Formulation

In this paper, we considered that each individual robot in the group carries sensors to exchange information within the multi-robot systems. As shown in Figure 1, assume at time $t_{1}$ that Robot b sends a request for communication and ranging to Robot a; after data transmission time $T_{a}$, Robot a receives the requests from Robot b at time $t_{2}$, and it takes time $T_{b}$ to process this information. Then, Robot a sends the relative distance and pose estimation to Robot b at time $t_{3}$, and Robot b receives the relative measurements from Robot a at time $t_{4}$ after data transmission time $T_{c}$. During the communication between Robot a and Robot b, the time delay we considered in this paper included data transmission time $T_{a}$, $T_{c}$, and the information processing time $T_{b}$. In practical work, the data transmission time $T_{a}$ and $T_{c}$ are typically very short, therefore, this paper will pay more attention to the information processing time $T_{b}$. We considered the situation that the information processing time $T_{b}$ caused by communication device was usually fixed, then, the time-invariant delay was considered in this paper.

As previously stated, there are N filtering periods during the entire communication process. Let Z(k) be the measurements that Robot b received at time $t_{4}$, the relative range and absolute position information received from Robot a are used to update the measurements of the systems. Due to the existence of delay, this state vector used for measurement update is actually measured at time $t_{2}$. Therefore, the problem this paper dealt with is, was how to improve the location accuracy and reliability of the systems under the time-invariant delay.

### 2.1. The Augmented State Motion Model with Delay

Consider the augmented state model with time delay. For the convenience of analysis, the leader–follower structure was used for the multi-robot systems in this paper, and we use symbol $t_{s}$ and $t_{k}$ as substitutes for $t_{2}$ and $t_{4}$ in the following equations.

The states of the systems are denoted as
(1)$$X\left(k\right)={\left(x_{l}\left(s\right),y_{l}\left(s\right),x_{f}\left(k\right),y_{f}\left(k\right),\theta_{f}\left(k\right)\right)}^{T}$$
where the subscript l and f denote the leader robot and the follower robot, respectively; $x_{l}\left(s\right)$, $y_{l}\left(s\right)$ is the state of the leader robot at time $t_{s}$; $x_{f}\left(k\right)$, $y_{f}\left(k\right)$, $\theta_{f}\left(k\right)$ are the state of the follower robot at time $t_{k}$. Then, the linearized state equation of the follower robot is described as follows
(2)$${\tilde{X}}_{f}\left(k+1\right)=\Phi_{f}\left(k\right){\tilde{X}}_{f}\left(k\right)+\Gamma_{f}\left(k\right)\omega_{f}\left(k\right)$$
where ${\tilde{X}}_{f}\left(k\right)$ denotes the state of the follower robot in cooperative positioning, and
(3)$$\Phi_{f}\left(k\right)=I+\delta t\left[\begin{array}{ccc} 1 & 0 & -\delta tv_{f}\left(k\right)\sin\left(\theta_{f}\left(k\right)\right)\\ 0 & 1 & \delta tv_{f}\left(k\right)\cos\left(\theta_{f}\left(k\right)\right)\\ 0 & 0 & 1 \end{array}\right]$$
(4)$$\Gamma_{f}\left(k\right)=\left[\begin{array}{cc} \delta t\cos\left(\theta_{f}\left(k\right)\right) & 0\\ \delta t\sin\left(\theta_{f}\left(k\right)\right) & 0\\ 0 & \delta t \end{array}\right]$$
and δt is the period time of sampling; $v_{f}\left(k\right)$ and $\theta_{f}\left(k\right)$ are the linear and rotational velocity of the robot at time $t_{k}$, respectively. $\omega_{f}\left(k\right)$ is the system noise due to the errors in the linear and rotational velocity measurements of the follower robot. The system inputs are denoted as
(5)$$u\left(k\right)=\left(x_{l}\left(s\right),y_{l}\left(s\right),v_{f}\left(k\right),\theta_{f}\left(k\right)\right)$$

Then, the linearized augmented state equation can be described as
(6)$$\tilde{X}\left(k+1\right)=\Phi \left(k\right)\tilde{X}\left(k\right)+\Gamma \left(k\right)\omega \left(k\right)$$
where
(7)$$\left\{ \begin{array}{l} \tilde{X}\left(k+1\right)={\left[\begin{array}{cc} {\tilde{X}}_{l}{\left(s+1\right)}^{T} & {\tilde{X}}_{f}{\left(k+1\right)}^{T} \end{array}\right]}^{T}\\ \omega \left(k\right)={\left[\begin{array}{cc} \omega_{l}{\left(s\right)}^{T} & \omega_{f}{\left(k\right)}^{T} \end{array}\right]}^{T}\\ \Phi \left(k\right)=\mathrm{diag}\left(\begin{array}{cc} \Phi_{l}\left(s\right) & \Phi_{f}\left(k\right) \end{array}\right)\\ \Gamma \left(k\right)=\mathrm{diag}\left(\begin{array}{cc} \Gamma_{l}\left(s\right) & \Gamma_{f}\left(k\right) \end{array}\right) \end{array}\right.$$
and the system noise covariance is given by
(8)$$Q\left(k\right)=E\left[\omega \left(k\right)\;\omega^{T}\left(k\right)\right]=\left[\begin{array}{cc} \sigma_{v}^{2}\left(k\right) & 0\\ 0 & \sigma_{\theta }^{2}\left(k\right) \end{array}\right]$$
and Φ(k) and Γ(k) are the Jacobian matrices of the state vector and the error vector, respectively.

### 2.2. Measurement Model with Delay

Suppose that the follower robot receives the ranging information from the leader robot. The range measurement model with communication delay can be described shortly as
(9)Z(k)=C(k)X(k)+v(k)

According to the states defined in Equation (1), the states of the leader robot at time $t_{s}$ and the states of the follower robot at time $t_{k}$ are included in the states of the leader–follower robot system. However, the measurement of the system only consists of the self-location of the leader robot and the relative position information at time $t_{s}$. Consider that the measurement at $t_{k}$ equals the measurement at $t_{s}$ after N filtering periods. Then, the measurement model can be reconstructed as
(10)$$\begin{array}{ll} Z\left(k\right) & =C\left(s\right)X\left(s\right)+v\left(k\right)\\ & =c_{l}\left(s\right)X_{l}\left(s\right)+c_{f}\left(s\right)X_{f}\left(s\right)+v\left(k\right) \end{array}$$

By utilizing the system state transition matrix, we have
(11)$$\begin{array}{ll} X_{f}\left(s+1\right) & =\vartheta \left(s+1,\;k+1\right)X_{f}\left(k+1\right)\\ & =\vartheta {\left(k+1,\;s+1\right)}^{-1}X_{f}\left(k+1\right) \end{array}$$
where the relationship of the one-step prediction state of the follower robot at time $t_{k}$ and the system state at time $t_{s}$ is described in Equation (10). Substituting Equation (10) in Equation (9) yields:(12)$$\begin{array}{l} Z\left(k\right)=c_{l}\left(s\right)X_{l}\left(s\right)+c_{f}\left(s\right)X_{f}\left(s\right)+v\left(k\right)\\ \;=c_{l}\left(s\right)X_{l}\left(s\right)+\vartheta_{f}{\left(k,s\right)}^{-1}X_{f}\left(k\right)+v\left(k\right)\\ \;=\left[c_{l}\left(s\right)\;c_{f}\left(s\right)\vartheta_{f}{\left(k,s\right)}^{-1}\right]\left[\begin{array}{c} X_{l}\left(s\right)\\ X_{f}\left(k\right) \end{array}\right]+v\left(k\right)\\ \;=C\left(k\right)X\left(k\right)+v\left(k\right) \end{array}$$

Then, the equivalent measurement equation of the system at time $t_{k}$ is deduced from the above formula. The system state transition matrix is given as
(13)$$\vartheta \left(k,s\right)=\prod_{i=s}^{k}\vartheta \left(i\right)$$
(14)$$\vartheta_{f}{\left(k,s\right)}^{-1}=\prod_{i=s}^{k}\vartheta_{f}{\left(i\right)}^{-1}=\left[\begin{array}{cc} 1 & -\sum_{i=s}^{k}L\left(i\right)\\ 0 & 1 \end{array}\right]$$
(15)$$L\left(i\right)=\left[\begin{array}{c} -\delta t^{2}V_{f}\left(i\right)\sin\left(\theta_{f}\left(i\right)\right)\\ -\delta t^{2}V_{f}\left(i\right)\cos\left(\theta_{f}\left(i\right)\right) \end{array}\right]$$

Then, the measurement equation of the multi-robot systems with time delay can be rewritten as
(16)Z(k)=C(k)X(k)+v(k)
where
$$\left\{ \begin{array}{l} C\left(k\right)=[c_{l}(s)\;c_{f}(k)]\\ c_{f}(k)=c_{f}\left(s\right)\vartheta_{f}{\left(k,s\right)}^{-1}\\ X\left(k\right)={\left[\begin{array}{cc} X_{l}{\left(s\right)}^{T} & X_{f}{\left(k\right)}^{T} \end{array}\right]}^{T} \end{array}\right.$$
$$c_{l}\left(s\right)=\frac{\partial Z\left(s\right)}{\partial X_{l}{\left(s,s-1\right)}^{T}},\;c_{f}\left(s\right)=\frac{\partial Z\left(s\right)}{\partial X_{f}{\left(s,s-1\right)}^{T}}$$
and the covariance matrix of the measurement noise is
(17)$$R\left(k\right)=E\left[v\left(k\right)\;v^{T}\left(k\right)\right]$$

## 3. Cooperative Localization with Communication Delays

The system and measurement model with communication delay has been expressed in the previous section. Based on the measurement update, this paper proposed a delayed extended Kalman filter (DEKF) to deal with the problem of cooperative localization with communication delays.

Considering that Z(k) is the equivalent measurement generated by the measurement information sent by the leader robot at time $t_{s}$ after N filtering cycles. In this section, the equivalent measurement at time $t_{k}$ is introduced into the measurement equation, and the optimal estimation of the system is obtained by using the principle of minimum variance estimation of error.

First, the one-step prediction state of the multi-robot system is given as follows:(18)$$\overset{⌢}{X}\left(k+1,k\right)={\left[\begin{array}{cc} {\overset{⌢}{X}}_{l}{\left(s+1,s\right)}^{T} & {\overset{⌢}{X}}_{f}{\left(k+1,k\right)}^{T} \end{array}\right]}^{T}$$
where ${\overset{⌢}{X}}_{l}{\left(s+1,s\right)}^{T}$ and ${\overset{⌢}{X}}_{f}{\left(k+1,k\right)}^{T}$ denote the one-step prediction states of the leader robot at time $t_{s}$ and the leader robot at time $t_{k}$, respectively.

Based on the linearized augmented state models (4), the one-step state prediction is given as follows:(19)$$\overset{⌢}{X}\left(k+1,k\right)=\Phi \left(k\right)\overset{⌢}{X}\left(k\right)$$
and its covariance are also given as follows:(20)$$P\left(k+1,k\right)=\Phi \left(k\right)P\left(k\right)\Phi^{T}\left(k\right)+\Gamma \left(k\right)Q\left(k\right)\Gamma^{T}\left(k\right)$$

The estimative state of the system is
(21)$$\overset{⌢}{X}\left(k\right)=\overset{⌢}{X}\left(k,k-1\right)+K\left(k\right)\left(Z\left(k\right)-C\left(k-1\right)\overset{⌢}{X}\left(k,k-1\right)\right)$$
where K(k) is an arbitrary filter gain, which requires minimum error estimation. The value of K(k) is determined according to the following formula:(22)$$\frac{\partial \mathrm{trace}(P(k))}{\partial K(k)}=0$$

If the possibility of communication delays is not a consideration, the general KF algorithm can be used to deduce the error prediction as follows:(23)$$\begin{array}{l} {\overset{⌢}{X}}^{\prime }\left(k,k-1\right)=\Phi \left(k-1\right)\overset{⌢}{X}\left(k-1\right)\\ =\Phi \left(k-1\right)\left[\begin{array}{l} \left(I-K^{\prime }\left(k-1\right)C\left(k-1\right)\right)\overset{⌢}{X}\left(k-1,k-2\right)\\ +K^{\prime }\left(k-1\right)Z\left(k-1\right) \end{array}\right]\\ =\Phi \left(k-1\right)\left[\begin{array}{l} \left(I-K^{\prime }\left(k-1\right)C\left(k-1\right)\right)\Phi \left(k-2\right)\overset{⌢}{X}\left(k-2\right)\\ +K^{\prime }\left(k-1\right)Z\left(k-1\right) \end{array}\right]\\ =\left[\prod_{i=1}^{N-1}\Phi \left(k-i\right)\left(I-K^{\prime }\left(k-i\right)C\left(k-i\right)\right)\right]\\ \;\cdot \Phi \left(s\right)\left[\prod_{i=1}^{N-1}{\overset{⌢}{X}}^{\prime }\left(s,s-1\right)+K^{\prime }\left(s\right){\overset{⌢}{X}}^{\prime }\left(s,s-1\right)\right]\\ \;+\sum_{j=3}^{N}\left[\begin{array}{l} \prod_{i=1}^{j-2}\Phi \left(k-i\right)\left(I-K^{\prime }\left(k-i\right)C\left(k-i\right)\right)\\ \;\cdot \Phi \left(k-j+1\right)K^{\prime }\left(k-j+1\right)Z\left(k-j+1\right) \end{array}\right]\\ \;+\Phi \left(k-1\right)K^{\prime }\left(k-1\right)Z\left(k-1\right) \end{array}$$

The above formula describes the correspondence between one-step predictive state at time $t_{k}$ and the estimation states among N Kalman filters.

When communication delays are considered, that I,s the measurement is unknown at time $t_{s}$. Then, the above formula can be rewritten as
(24)$$\begin{array}{l} \overset{⌢}{X}\left(k,k-1\right)\\ =\left[\prod_{i=1}^{N-1}\Phi \left(k-i\right)\left(I-K\left(k-i\right)C\left(k-i\right)\right)\right]\Phi \left(s\right)\overset{⌢}{X}\left(s,s-1\right)\\ \;+\sum_{j=3}^{N}\left[\begin{array}{l} \prod_{i=1}^{j-2}\Phi \left(k-i\right)\left(I-K\left(k-i\right)C\left(k-i\right)\right)\\ \;\cdot \Phi \left(k-j+1\right)K\left(k-j+1\right)Z\left(k-j+1\right) \end{array}\right]\\ \;+\Phi \left(k-1\right)K\left(k-1\right)Z\left(k-1\right) \end{array}$$

Then, the error compensation of the one-step predictive state at time $t_{k}$ can be obtained as follows:(25)$$\begin{array}{l} \Delta \overset{⌢}{X}\left(k,k-1\right)={\overset{⌢}{X}}^{\prime }\left(k,k-1\right)-\overset{⌢}{X}\left(k,k-1\right)\\ =\left[\prod_{i=1}^{N-1}\Phi \left(k-i\right)\left(I-K^{\prime }\left(k-i\right)C\left(k-i\right)\right)\right]\Phi \left(s\right)\left[\overset{⌢}{X}\left(s,s-1\right)+K^{\prime }\left(s\right)\left(Z\left(s\right)-C\left(s\right)\overset{⌢}{X}\left(s,s-1\right)\right)\right]\\ \;-\left[\prod_{i=1}^{N-1}\Phi \left(k-i\right)\left(I-K\left(k-i\right)C\left(k-i\right)\right)\right]\Phi \left(s\right)\overset{⌢}{X}\left(s,s-1\right)\\ \;+\sum_{j=3}^{N}\left[\prod_{i=1}^{j-2}\Phi \left(k-i\right)\left(I-K^{\prime }\left(k-i\right)C\left(k-i\right)\right)\Phi \left(k-j+1\right)K^{\prime }\left(k-j+1\right)Z\left(k-j+1\right)\right]\\ \;-\sum_{j=3}^{N}\left[\prod_{i=1}^{j-2}\Phi \left(k-i\right)\left(I-K\left(k-i\right)C\left(k-i\right)\right)\Phi \left(k-j+1\right)K\left(k-j+1\right)Z\left(k-j+1\right)\right]\\ \;+\Phi \left(k-1\right)K^{\prime }\left(k-1\right)Z\left(k-1\right)-\Phi \left(k-1\right)K\left(k-1\right)Z\left(k-1\right) \end{array}$$

Note that K in Equation (25) is different to $K^{\prime }$, which is the filter gain without communication delays. $K^{\prime }$ is the filter gain without considering the system delay condition. K is the filter gain considering the system delay condition, and the difference between $K^{\prime }$ and K is whether the measurement information is added into the system at time $t_{s}$. At time $t_{s}$, the measurement matrix and the measurement noise covariance matrix can be estimated, then Equation (25) can be rewritten as
(26)$$\begin{array}{l} \Delta \overset{⌢}{X}\left(k,k-1\right)\\ =\left[\prod_{i=1}^{N-1}\Phi \left(k-i\right)\left(I-K\left(k-i\right)C\left(k-i\right)\right)\right]\Phi \left(s\right)M\left(s\right) \end{array}$$
where
(27)$$M\left(s\right)=K\left(s\right)\left(Z\left(s\right)-C\left(s\right)\overset{⌢}{X}\left(s,s-1\right)\right)$$

It can be seen from the above formula that the state compensation includes the system state matrix Φ(k−i) and the system state estimation error M(s) of the N filter periods from $t_{s}$ to $t_{k}$.

Therefore, the error compensation formula of the one-step predictive state at time $t_{k}$ can be written as
(28)$$X\left(k,k-1\right)=X^{\prime }\left(k,k-1\right)+\Delta \overset{⌢}{X}\left(k,k-1\right)$$

Finally, the cooperative localization algorithm for multi-robot systems can be summarized as follows:

**Algorithm 1:** The cooperative localization algorithm based on State Estimation Error Compensation1:**Initialize****:** Assume that each robot in the system initially knows its pose with respect to a given reference coordinate frame. As Figure 1 shows, consider that at time tk, the follower robot receives the pose information from the leader robot with time delay after N Kalman filters at time ts.2:  **State prediction and compensation:** Give the one-step state prediction and covariance matrix:X⌢(k+1,k)=Φ(k)X⌢(k)
P(k+1,k)=Φ(k)P(k)ΦT(k)+Γ(k)Q(k)ΓT(k)3:  Calculate the state estimation error compensation:M(k)=Φ(k)(I−K(k)C(k))M(k−1)
ΔX⌢(k+1,k)=M(k)Φ(s)K(s)(Z(s)−C(s)X⌢(s−1,s))4:  Compute the filter gain:K(k+1)=P(k+1,k)CT(k)[C(k)P(k+1,k)CT(k)+R(k)]−15:  Construct the error-state propagation equation and the covariance propagation equation:X⌢(k+1,k)=X′(k+1,k)+ΔX⌢(k+1,k)
X⌢(k+1)=X⌢(k+1,k)+K(k+1)(Z(k+1)−C(k)X⌢(k+1,k))
P(k+1)=(I−K(k+1)C(k+1))P(k+1,k)6:**end**

With the cooperative localization algorithm proposed above, the extended Kalman filtering based on state estimation error compensation with communication delays can be designed to improve the location accuracy.

## 4. Simulation Analysis

### 4.1. Setup

In this section, the performance of the proposed algorithm was evaluated by a group of two robots with a leader–follower structure as shown in Figure 2. The group of robots moved in a rectangular area in an indoor environment. Both robot carried an orientable range finder and wheel encoders for odometry. The range finder was used to compute the measurements aimed at the leader robot. Both the range measurements and the odometric measurements were supposed to be affected by a zero-mean white Gaussian noise.

### 4.2. Results

In the simulations, it was assumed that the mean velocity of the robots was 1 m/s and the system noise and measurement noise covariance matrices used were as follows:(29)$$Q=\left[\begin{array}{cc} {\left(0.1m/s\right)}^{2} & 0\\ 0 & {\left(3^{0}\right)}^{2} \end{array}\right]$$
(30)$$R=\left[\begin{array}{cc} {\left(0.5m/s\right)}^{2} & 0\\ 0 & {\left(0.5^{0}\right)}^{2} \end{array}\right]$$

Communication delays are supposed to be time invariant such as 0.1 s. The filtering period was 0.8 s. As shown in Figure 3, the true trajectory for the leader–follower robot team was compared to the trajectory estimated using the proposed cooperative localization algorithm.

Figure 4 shows the local enlarged drawing of true trajectory and estimated trajectory. From the above simulation results, the effectiveness of the proposed method were verified.

Figure 5, Figure 6 and Figure 7 illustrate the comparative error curves of the follower robot by using the traditional EKF method in the first 100 steps and the proposed method in the next 200 steps, respectively. It can be seen that the estimation errors of the proposed algorithm were bounded in a smaller range when compared with the traditional EKF method.

In order to verify the computational efficiency of the DEKF filtering algorithm, Figure 8 shows the computation time of each step when the DEKF method and traditional EKF method were used for cooperative positioning.

It can be seen from Figure 8 that the computational efficiency of the DEKF algorithm was better than traditional EKF algorithm even though the time delay was added to the filtering process.

With the continuous movement of the multi-robot system, more and more information needs to be computed, but only the nearest state information can be used to locate the robot itself, and all the original information does not need to be saved. Therefore, a limited storage capacity can be maintained by deleting unnecessary storage content.

## 5. Conclusions

In this paper, the delayed cooperative localization problem for a leader–follower robot system was considered. An extended Kalman filter approach based on state estimation error compensation was proposed to keep positioning accuracy with communication delays. The simulation and experiment results demonstrated that the estimation accuracy of the proposed method was comparable with the traditional EKF localization method. However, this method must preserve the state of the system during the entire period of the delay. If the delay is long, it may impose high requirements on the system storage hardware. In addition, if there are data loss, it will affect the algorithm. How to reduce the amount of storage and deal with the loss of packets without the loss of precision is the next topic to be studied.

## Figures and Tables

**Figure 1 sensors-19-03842-f001:**
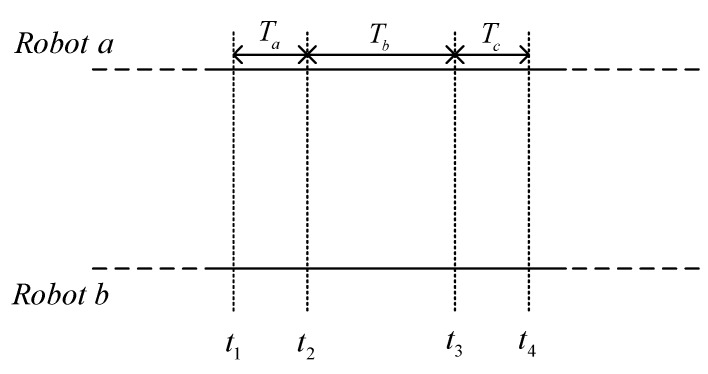
Timeline of the robot-to-robot communication.

**Figure 2 sensors-19-03842-f002:**
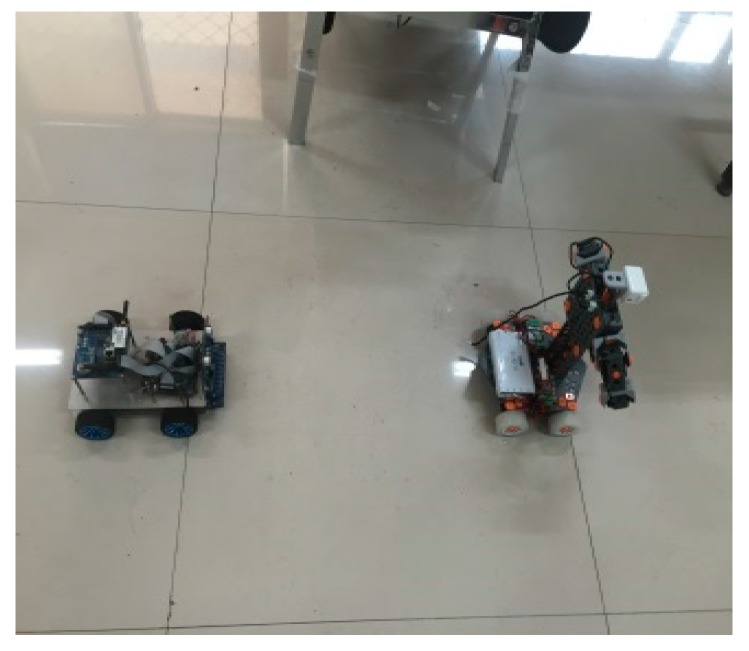
Experimental setup used for testing the proposed algorithm.

**Figure 3 sensors-19-03842-f003:**
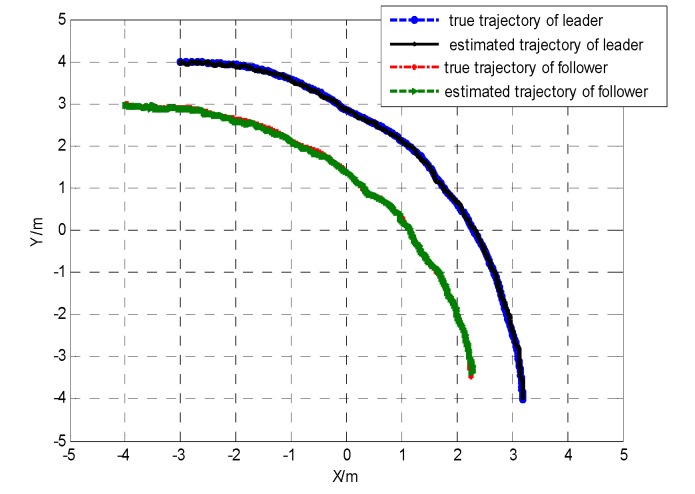
The true trajectory and the estimated trajectory for the leader–follower robot team.

**Figure 4 sensors-19-03842-f004:**
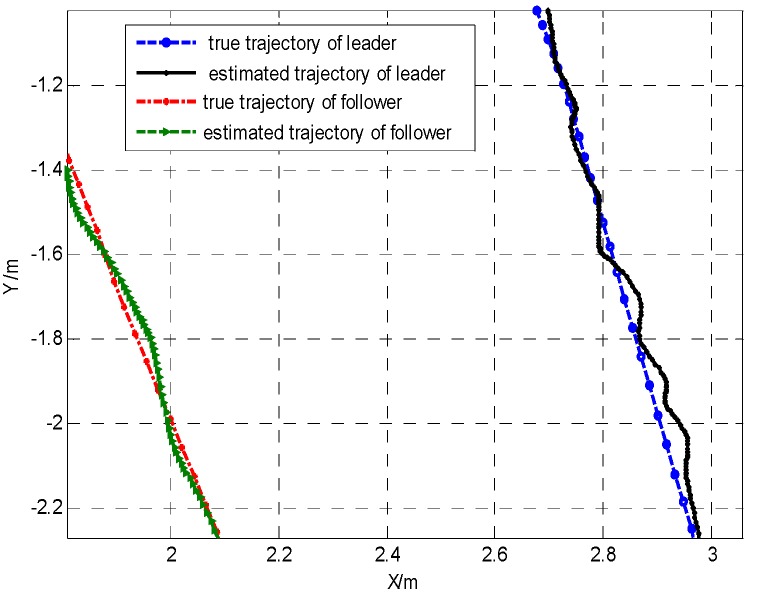
The local enlarged true trajectory and the estimated trajectory for the leader–follower robot team.

**Figure 5 sensors-19-03842-f005:**
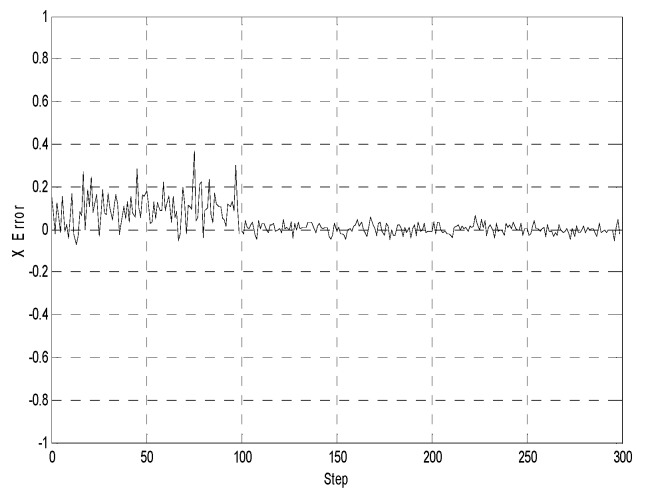
X-position estimates (the first 100 steps using the extended Kalman filtering method without considering the communication delays, the next 200 steps using the proposed method).

**Figure 6 sensors-19-03842-f006:**
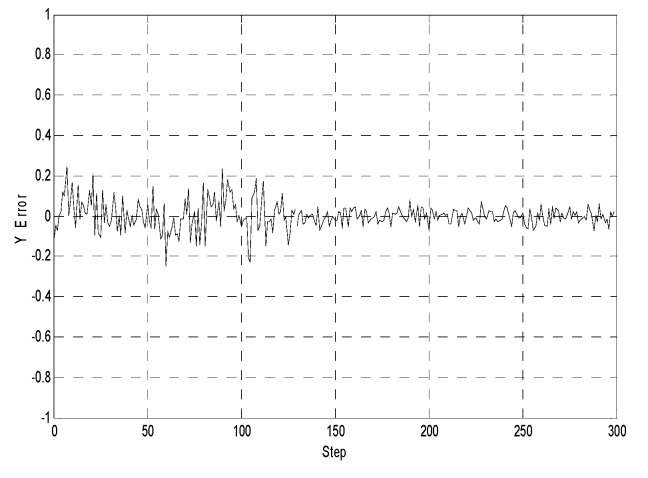
Y-position estimates (the first 100 steps using the extended Kalman filtering method without considering the communication delays, the next 200 steps using the proposed method).

**Figure 7 sensors-19-03842-f007:**
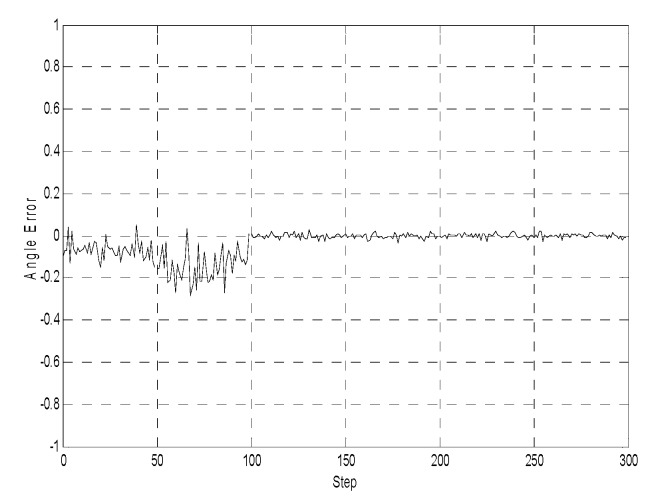
Orientation estimates (the first 100 steps using the EKF method without considering the communication delays, the next 200 steps using the proposed method).

**Figure 8 sensors-19-03842-f008:**
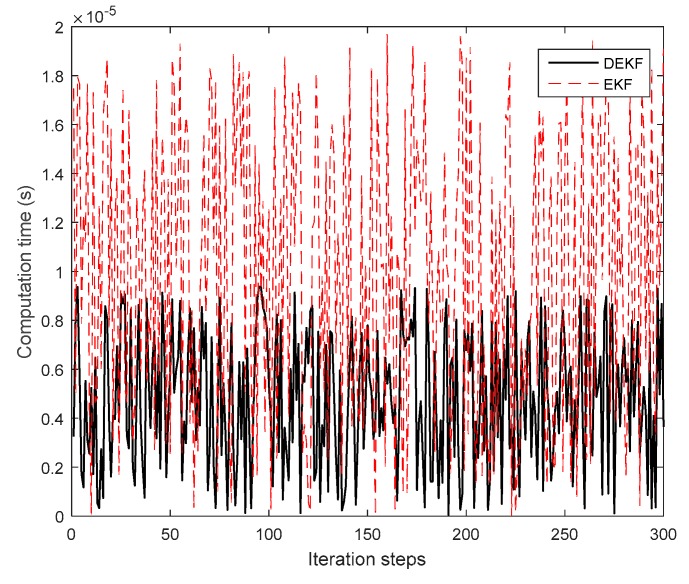
The computation time of each step.

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
