# Peer review of "Cooperative Localization Approach for Multi-Robot Systems Based on State Estimation Error Compensation"

_sensors, 2019, doi:10.3390/s19183842_

Round 1
Reviewer 1 Report
This paper describes an interesting modeling approach applied to robotics. The topic is attractive. The theory is presented at the minimum understandable level and the application is trying to support the theory. The paper requires efforts improvements using the following comments:
1) The main contribution must be specified.
2) The motivation of the approach must also be specified in the context of other similar modeling approaches.
3) Discussion of related work on linear and nonlinear modeling approaches should be extended with the following papers, which recently came into my attention as they proved to be successful in quite different applications:
* Predictive functional control based on fuzzy model: design and stability study, Journal of Intelligent and Robotic Systems, vol. 43, no. 2-4, pp. 283-299, 2005.
* Neural network based feature extraction for assamese character and numeral recognition, International Journal of Artificial Intelligence, vol. 2, no. S9, pp. 37-56, 2009.
* New results in modelling derived from Bayesian filtering, Knowledge-Based Systems, vol. 23, no. 2, pp. 182-194, 2010.
* Model-free sliding mode and fuzzy controllers for reverse osmosis desalination plants, International Journal of Artificial Intelligence, vol. 16, no. 2, pp. 208-222, 2018.
4) The paper contains some grammar problems. Their correction is needed.
5) You should also correct and explain the notations especially those with two arguments as theta and X.
Moreover, you are using theta with one argument and theta with two arguments. This might be confusing.
6) The initial conditions are not specified.
7) The time delay should be visible.
8) How do you compensate for the effects of the time delay?
9) You should specify how did you compute the parameters of EKF.
10) A better connection of Section 4 to the previous sections is necessary in order to show how the theory was effectively applied here.
11) You could save the programs and schemes in a webpage and cite the link to that webpage in the paper body. That would ensure a sound validation to be checked by the readers.
12) A comparison with a similar approach would be useful.
All in all, I appreciate it and do hope that these comments will improve it.
Reviewer 2 Report
# Summary
The paper is well written from a grammatical point of view. Up to my knowledge, the bibliography seems essential, but other references can be found in papers cited by the authors. Some suggestions are reported at the end of this review.
This paper faces the localization problem for multi-robot systems in the presence of communication delay.
The authors assumed that each robot is able to exchange information with other robots but delays appear in the communication. Consider two robots A and B. The first delay appears when robot A sends its information, a second delay appears because robot B has to process the data received and finally, a third source of delay comes from sending the information from robot B to robot A.
The authors modeled the robot as a leader-follower model, where the state of this system is the position of the leader and the pose (position and orientation) of the follower. The authors then linearize this dynamics and introduced the measurement model that includes the delay. Roughly speaking, it seems to me that the model presented by the authors only propagates the same measurements in case there is delay in the communication. Th authors then introduced the Extended Kalman Filter (EKF) in presence of delay. The major difference w.r.t. the EKF that does not includes the communication delay relies in the second line of eq.(14). Finally, the authors introduced their algorithm, that effectively summarizes the paper.
Regarding the experimental part, the authors proposed a scenario where the robot moves at constant speed and the communication delay is constant. The paper ends with a comparison between the proposed approach and the EKF that does not consider communication delay, showing the effectiveness of the proposed approach.
# Consideration about contribution
The main contribution of this paper relies in the inclusion of the delay inside a leader-follower dynamical system. I particularly appreciated Algorithm 1, since it effectively summarizes all the paper. However, I have some doubts about the paper:
1) In eq.(3), I did not get why the position of the leader is considered as an input of the system.
2) The major drawback of the paper relies in the contribution. This is a journal article and the contribution, in my humble opinion, should be stronger. In fact, equations apart, the authors proposed an algorithm that propagates the measurements in case of communication delay. For example, it would be interesting to provide how the proposed framework can be extended for multiple robots. For example, is it possible to consider two or more-HOP communications (i.e., consider three robot: robot A,B,C. Robot C is able to receive the information from robot A just through robot B. In other words, robot C does not have a direct channel with robot A and it is able to receive its information in two steps.)?
3) The simulations performed can be improved. The authors focused on the simplest scenario: two robots and constant time delay. How the framework performs in presence of variable delay? This should be interesting because the framework does not seem to estimate the communication delay and there is no clue that the framework will work in such a scenario.
# Minor comments
1) pag 1: 1980, in multi-robot systems -> delete "in multi-robot systems" since the authors already used it at the beginning of the sentence.
2) pag 1: the bibliography style should be revised. Sometimes, the citations are in subscript, sometimes no.
3) pag 2: Consider the reason above -> Considering the reasons above
4) pag 2: Robot b sends communication. I did not get the sense of this. Is "the robot sends some data"? In case, which data?
5) pag 2: "In the entire communication process...". I completely did not get the meaning of this sentence.
6) pag 2: Denote Z(k) is -> Let Z(k) be
7) pag 3: these state vector -> this state vector
7) pag 3: that how -> delete "that"
8) pag 3: are the state -> is the state
9) pag 4: only consist -> only consists
10) pag 4: are included in the state of system. Which system? The overall (leader+follower) system?
11) pag 8: gauss noise -> Gaussian noise
12) pag 9: I did not get the differences between Figures 3 and 4. Is Figure 4 a zoom of Figure 3?
# Bibliography suggestions
1) X. Shenet al. A General Framework for Multi-vehicle Cooperative Localization Using Pose Graph. arXiv:1704.01252
2) M. Cognetti et al. Real-Time Pursuit-Evasion with Humanoid Robots. In 2017 IEEE International Conference on Robotics and Automation (ICRA), Pages 4090-4095, May 2017
2) M. D. Phung et al. Localization of networked robot systems subject to random delay and packet loss. 2013 IEEE/ASME International Conference on Advanced Intelligent Mechatronics, 1442-1447
3) M. Cognetti et al. Optimal Active Sensing with Process and Measurement Noise. In IEEE Int. Conf. on Robotics and Automation, ICRA'18, Pages 2118-2125, Brisbane, Australia, May 2018
4) P. Stegagno et. al. Ground and Aerial Mutual Localization Using Anonymous Relative-Bearing Measurements. IEEE Transactions on Robotics, 32(5):1133-1151, October 2016
Reviewer 3 Report
The submitted manuscript presents a cooperative localization approach for multi-robot systems, which employs state estimation error compensation. Finally, the method is experimentally tested in a laboratory environment. In the reviewer’s opinion there are still some concerns or questions which should be addressed by the authors before its acceptance:
(1)In this paper, a novel localization approach for multi-robot systems is proposed. The benefits of the method have been illustrated clearly. Are there other ways that the assumption of the condition can be further reduced?
(2) Information fusion provides a powerful tool to deal with uncertainty and external disturbance. For example, Human-Manipulator Interface based on Multisensory Process via Kalman Filters, A Markerless Human-Robot Interface Using Particle Filter and Kalman Filter for Dual Robots, Markerless Human-Manipulator Interface Using Leap Motion with Interval Kalman Filter and Improved Particle Filter. Brief discussions are helpful. The reviewer doubts if the authors can combine the sliding mode estimation technique to improve the robustness of the developed method.
(3) In conclusion part, more future works and challenges are recommended.
Round 2
Reviewer 1 Report
Authors extended and improved their paper. It is acceptable for publication in this prestigious journal. Please just take care to authors' names in References section and use all instead of "et al.".
Author Response
1. Authors extended and improved their paper. It is acceptable for publication in this prestigious journal. Please just take care to authors' names in References section and use all instead of "et al.".
We have have added the names of all authors in References section instead of "et al."
We deeply appreciate your constructive comments that greatly help
improve the technical quality and the presentation of this manuscript.
Reviewer 3 Report
After revision, some issues have been addressed and the paper has been significantly improved. In the reviewer’s opinion there are still some concerns or questions which should be addressed by the authors before its acceptance:
(1) Experimental results can be described in more detail.
(2) The introduction can go further.
Author Response
(1) Experimental results can be described in more detail.
In Section 4, in order to verify the computational efficiency of DEKF filtering algorithm,we discuss the computational efficiency of the proposed method and compare it with the traditional EKF. It can be seen from Fig8, the computational efficiency of the DEKF algorithm is better than traditional EKF algorithm although the time delay is added to the filtering process.
(2) The introduction can go further.
In Section 1,more discussion about Kalman filters which can deal with uncertainty and external disturbance in the measurement have been joined,and more may refer to the references[17-19].